# Utility and Mechanism of SHetA2 and Paclitaxel for Treatment of Endometrial Cancer

**DOI:** 10.3390/cancers13102322

**Published:** 2021-05-12

**Authors:** Vishal Chandra, Rajani Rai, Doris Mangiaracina Benbrook

**Affiliations:** Gynecologic Oncology Section, Stephenson Cancer Center, Department of Obstetrics and Gynecology, University of Oklahoma Health Sciences Center, Oklahoma City, OK 73104, USA; Vishal-Chandra@ouhsc.edu (V.C.); Rajani-Rai@ouhsc.edu (R.R.)

**Keywords:** endometrial cancer, SHetA2, paclitaxel, cell cycle arrest, mitochondria, metabolism, oxidative phosphorylation, glycolysis, apoptosis inducing factor, xenograft

## Abstract

**Simple Summary:**

Incidence and death rates for endometrial cancer are steadily rising world-wide. Endometrial cancer patients at high risk for recurrence are treated with chemotherapy, which causes significant toxicity. Molecularly targeted drugs have been found to cause less toxicity than chemotherapy. We studied a low-toxicity drug, called SHetA2, which targets three heat shock A proteins that are highly mutated in endometrial cancers. Our results demonstrated that SHetA2 inhibits endometrial cancer cells and tumors, and enhances therapeutic effects of paclitaxel without increasing toxicity. This information supports development of clinical trials to test if combining SHetA2 with paclitaxel can increase the paclitaxel therapeutic effect without increasing toxicity, or allows a lowered paclitaxel dose to achieve the same level of therapeutic effect, but with reduced toxicity. Our new knowledge about how SHetA2 works can be translated into development of biomarkers to predict with patients would most likely benefit from SHetA2-based therapy.

**Abstract:**

Endometrial cancer patients with advanced disease or high recurrence risk are treated with chemotherapy. Our objective was to evaluate the utility and mechanism of a novel drug, SHetA2, alone and in combination with paclitaxel, in endometrial cancer. SHetA2 targets the HSPA chaperone proteins, Grp78, hsc70, and mortalin, which have high mutation rates in endometrial cancer. SHetA2 effects on cancerous phenotypes, mitochondria, metabolism, protein expression, mortalin/client protein complexes, and cell death were evaluated in AN3CA, Hec13b, and Ishikawa endometrial cancer cell lines, and on growth of Ishikawa xenografts. In all three cell lines, SHetA2 inhibited anchorage-independent growth, migration, invasion, and ATP production, and induced G1 cell cycle arrest, mitochondrial damage, and caspase- and apoptosis inducing factor (AIF)-mediated apoptosis. These effects were associated with altered levels of proteins involved in cell cycle regulation, mitochondrial function, protein synthesis, endoplasmic reticulum stress, and metabolism; disruption of mortalin complexes with mitochondrial and metabolism proteins; and inhibition of oxidative phosphorylation and glycolysis. SHetA2 and paclitaxel exhibited synergistic combination indices in all cell lines and exerted greater xenograft tumor growth inhibition than either drug alone. SHetA2 is active against endometrial cancer cell lines in culture and in vivo and acts synergistically with paclitaxel.

## 1. Introduction

World-wide, the incidence and mortality rates for endometrial cancer have been steadily rising over the past two decades, in contrast to decreases in these rates for most other cancers [1,2]. The increasing world-wide epidemic of obesity is implicated in the higher risk for developing endometrial cancer and the higher all-cause mortality of endometrial cancer patients [3,4,5]. While strategies to combat obesity are making efforts to stem this rising tide of endometrial cancer in the long-term, development of new therapeutic strategies have the potential for having a more immediate impact on endometrial cancer patient outcomes. Molecular characterization of endometrial tumors offers an opportunity to identify candidate targets for development of new therapeutic options.

The glucose regulated protein 78 (Grp78), Hsc70, and mortalin proteins encoded by the *heat shock protein A* (*HSPA*) *5*, *8*, and *9* genes, respectively, have the highest mutation frequencies in endometrial cancer compared to other cancers evaluated in The Cancer Genome Atlas (TCGA) [6]. Grp78, hsc70, and mortalin are molecular chaperones that play important roles in cell survival by binding and assuring proper folding, function, and cellular localization of their client proteins and protein complexes. Grp78 is the key gatekeeper for detecting endoplasmic reticulum (ER) stress and activating the unfolded protein response (UPR) [7]. In endometrial cancer, ER stress and Grp78 expression are elevated, and increased Grp78 expression is associated with platinum and taxane resistance [8,9,10]. hsc70 is the sole chaperone known to bring old and damaged proteins directly to the lysosome for recycling in chaperone mediated autophagy [11]. Inhibiting autophagic degradation of proteins can increase response of endometrial cancer cell lines to molecularly-targeted agents both in vitro and in vivo [12]. Although mortalin has many functions throughout the cell, it primarily localizes to mitochondria where it plays an essential role in importing calcium and nuclear encoded proteins, and maintenance of mitochondrial integrity [13]. Elevated mortalin expression is needed to maintain immortalization [14,15] and can drive carcinogenesis [16], while its further elevation in cancer cells is associated with patient response to therapy and survival [17,18,19,20,21,22,23,24].

We developed the drug, Sulfur Heteroarotinoid A2 (SHetA2), which binds Grp78, Hsc70, and mortalin proteins and disrupts their binding to client proteins [17,25]. SHetA2 is a novel small molecule, orally-bioavailable drug entering NCI R01-supported Phase 1 clinical trials in ovarian, cervical, and endometrial cancers. Extensive preclinical studies conducted by the NCI, the authors, and others showed that SHetA2 does not cause toxicity, cardiac effects, skin irritancy, teratogenicity, or mutagenicity [26,27,28,29,30]. The No Observed Adverse Effect Level of 28-day oral SHetA2 in dogs is 1,500 mg/kg/day [30], which represents a very large therapeutic window compared to the 10–60 mg/kg/day oral SHetA2 dose range shown to reduce in vivo tumor development and growth [17,26,29,31,32]. Pharmacokinetic and metabolic studies documented that SHetA2 is not a pro-drug, and that effective SHetA2 concentrations can be achieved in cancer tissues with oral dosing of 60 mg/kg/day SHetA2 suspended in Kolliphor HS15 [33,34,35,36,37]. This formulation was used in development and current Good Manufacturing Production (cGMP) of SHetA2 capsules for the clinical trials [38].

Current directions in development of new treatment strategies involve testing drug drug combinations with complementary mechanisms of action targeting specific defects in cancers [39]. A rational SHetA2-based combination therapy is adding SHetA2 to paclitaxel, which is a standard-of-care chemotherapeutic used to treat patients with endometrial cancer that is advanced stage or early stage at high risk for recurrence after surgery [40]. A major cause of paclitaxel resistance is centrosome duplication in the presence of paclitaxel-damaged chromosomes [41]. We predict that SHetA2 will increase paclitaxel sensitivity by releasing the mortalin client proteins, p53 and monopolar spindle 1 kinase (Mps1), which inhibit and promote centrosome duplication, respectively [42,43]. Mortalin super-activates Mps-1 to induce centrosome duplication [44]. Elevated mortalin can represses p53 at the centrosome and allow aberrant centrosome duplication [45]. Thus, SHetA2 could repress chromosome duplication in paclitaxel treated cells by disruption of mortalin/p53 complexes, thereby allowing p53 to accumulate in the nucleus, as previously demonstrated [17], and inhibit chrosome duplication. Simultaneously, SHetA2 disruption of mortalin/Mps1 kinase complexes could prevent Mps1 promotion of centrosome duplication.

The objective of this study as to evaluate the utility and mechanism of SHetA2 as a single agent and in combination with paclitaxel for treatment of endometrial cancers.

## 2. Materials and Methods

### 2.1. Cell Lines, Culture Conditions, and Chemicals

The Ishikawa endometrial cancer cell line was purchased from Sigma (St Louis, MO, USA). AN3CA and Hec1B were obtained as a gift from Jie Wu Ph.D., University of Oklahoma Health Sciences Center, Oklahoma City, OK, 73104, USA. All the three cell lines were grown in EMEM media supplemented with 10% FBS, 0.1% Insulin and 1% penicillin/streptomycin and passaged less than 20 times for all experiments in this study. For the cell culture studies, SHetA2 was synthesized by K. Darrell Berlin at Oklahoma State University [46] and dissolved in dimethylsulfoxide (DMSO, Neta Scientific, Hainesport, NJ, USA). For controls, cell cultures were treated with an equal amount of DMSO (vehicle) present in the treated cultures. SHetA2 provided by the US National Cancer Institute RAPID Program was suspended in 30% Kolliphor HS 15 (Sigma Aldrich, Merck, Darmstadt, Germany) in water for use in the animal model. Untreated control animals were treated with the same volume of 30% Kolliphor HS 15 as used for the treated animals.

### 2.2. Colony Formation Assay

For the colony formation assay, endometrial cancer cells (2 × 10^3^ per well of 12-well plates) in EMEM medium containing 20% FBS with 0.3% agar (# 50101, SeaPlaque™ Agarose, Lonza, Basel, Switzerland) were seeded on top of 0.6% agar in the same medium. Approximately 100 µL of culture medium, with or without SHetA2, was added to the top layer, and the plates were incubated at 37 °C in a humidified atmosphere of 5% CO_2_. After three weeks of incubation, the colonies were counted using GelCount (Oxford Optronix Ltd., Milton, Abingdon OX14 4SA, UK).

### 2.3. Wound Healing Assay

The role of SHetA2 in endometrial cancer cell migration was assessed using the wound healing assay. In brief, confluent monolayers of endometrial cancer cells, grown in six-well plates, were scratched with a 10 µL pipette tip to make a “wound”. Plates were washed to remove cell debris to clear a cell-free zone, followed by treatment with SHetA2 (5 µM) or vehicle. Images of the wound area were captured at 0 and 24 h in Ishikawa and 0 and 48 h in AN3CA and Hec1B, using an inverted Nikon Eclipse Ni microscope.

### 2.4. Invasion Assay

Endometrial cancer cells (2.5 × 10^4^) were seeded onto Matrigel (# 356232 Corning Inc., Corning, NY, USA)-coated Transwell chambers (24-well insert, 8 µm pore size; # 353097, Corning Inc., Corning, NY, USA) in a serum-reduced condition (EMEM supplemented with 5% FBS). Then, the cells were treated SHetA2 and the basal chamber was supplemented with EMEM media (with 10% FBS). After 72 h of treatment, the non-migrated cells on the apical side of the chamber were gently scraped off, while the migrated cells at the basal side were fixed in 4% paraformaldehyde and methanol. The migrated cells were then stained with 0.5% crystal violet and their images were captured under an inverted microscope using a Nikon Eclipse Ni microscope. For quantification of the numbers of migrated cells, the invaded cells were trypsinized and counted using a cell counter (Countess™ II FL Automated Cell Counter, #AMQAF1000, ThermoFisher, Waltham, MA, USA).

### 2.5. Cell Cycle Assay

Endometrial cancer cells treated with 5 µM SHetA2 or vehicle for 24 h were trypsinized, collected and washed with PBS. The cell suspension was fixed in 100% ethanol and incubated on ice for 20 min. Afterwards, the fixed cells were stained with Propidium Iodide (PI) Staining solution (0.5 μg/mL RNase A, 50 μg/mL PI and 1% Triton X-100) and Flow cytometry was performed using a FACS Calibur Flow Cytometer (BD Biosciences, San Jose, CA, USA).

### 2.6. Western Blot Analysis

Whole-cell protein extracts of endometrial cancer cells (treated with SHetA2 or vehicle for desired times) were collected in mPER (#78501, ThermoFisher, Waltham, MA, USA) containing 1% phosphatase inhibitor cocktail (#4906845001 Sigma-Aldrich, St Louis, MO, USA) and 1% protease inhibitor cocktail (#5892791001, Sigma-Aldrich). The total protein concentration was estimated with the BCA Assay Kit (#23225, ThermoFisher, Waltham, MA, USA). Equal amounts of protein (25 μg) were separated on an SDS-PAGE (sodium dodecyl sulfate-polyacrylamide gel electrophoresis) gel and transferred to PVDF (polyvinylidene difluoride) membranes using a Trans-Blot^®^ Turbo™ Transfer System (BIO-RAD, Hercules, CA, USA). The membranes were blocked in 10% nonfat milk in Tris-buffered saline (TBS) containing 0.1% *v/v* Tween-20 with 0.1% Tween-20 (TBST) for 1 h at room temperature followed by incubation with primary antibodies in TBST with 4% bovine serum albumin (BSA) overnight. After washing, membranes were incubated in HRP-conjugated anti-rabbit (#7074, Cell Signaling Technology, Danvers, MA, USA, at 1:5000 dilution) or anti-mouse antibody (cst#7076, Cell Signaling Technology, Danvers, MA, USA, at 1:6000 dilution) for 45 min at room temperature. Proteins of interest were visualized with an electrochemical luminescence reagent (Clarity™ Western ECL Substrate) (#1705060S, BioRad, Hercules, CA, USA) using a ChemiDocTM Touch Imaging System (BioRad). Equal loading was verified by immunoblotting with α-tubulin, cyclophilin-B or GAPDH antibodies. The following primary antibodies were used. The antibodies purchased from Cell signaling technologies are as follows, Caspase-3 (#9662), PARP (#9542), p27 (#3686), p21 (#2947), Cyclin D1 (#2922), pDRP (#4494), DRP1 (#5391), MFF (#84580), OPA1 (#80471), MFN1 (#14739), MFN2 (#11925), γH2AX (#2577), Tom20 (#42406), GAPDH (#5174), α-Tubulin (#2125), Cyclophilin B (43603), mortalin (#3593), IP3R(#8568), P62 (#5114), and LC3-II (#12741). Other antibodies purchased from ThermoFisher (Waltham, MA, USA) are AIF (# MA5-15880), ALDH18A1 (# PA5-52546), CTPS (# PA5-65655), MDH1 (# PA5-55573), ECHS1 (# 11305-1-AP). All Western blots were repeated at least at two different time points and are provided in Appendix A.

### 2.7. Mitochondrial Membrane Potential (MMP) Assay

MMP were measured using an MMP assay kit (ab113850; Abcam, Cambridge, MA, USA), according to the manufacturer’s instructions. For this, endometrial cancer cells were seeded onto 96-well black plates ((#NC1463153, Perkin Elmer, Waltham, MA, USA) at a density of 1.5 × 10^4^ cells/well. After an overnight incubation, cells were treated with SHetA2 or vehicle for the next 24 h. Afterwards, cells were washed with 1× dilution buffer and incubated with 20 µM JC-1 fluorescent dye (5′,6′,-tetrachloro-1,1′,3,3′ tetraethyl-benzimidazolyl carbocyanine iodide) for 10 min at 37 °C followed by two times washing with 1× dilution buffer. The fluorescence intensities for J-aggregates and J-monomers was measured at excitation and emission wavelengths of 535 nm and 590 nm, and 475 nm and 530 nm, respectively. The ratios of J-aggregates and J-monomers were calculated.

### 2.8. MTT Cell Viability Assay

Endometrial cancer cells (4–6 × 10^4^ cells/well) were seeded in 96-well plates for 24 h and treated with various concentrations of SHetA2 for 24, 48, and 72 h. After the specific treatment times, 15 µL of MTT solution (#G4100, Promega Madison, WI, USA) was added. Following one-hour incubation at 37 °C, 100 µL of STOP solution was added and allowed to incubate overnight at 37 °C. Then, the optical density (OD) of the 96-well plates were measured at a wavelength of 570 and 620 nm using a Bio-Tek Synergy H1 Micro Plate Reader (BioTek, Winooski, VT, USA) and Gen5 2.09 Software (BioTek, Winooski, VT, USA). The fold effect was calculated by dividing the average OD of the treated cultures by the average corrected OD of control cultures. GraphPad Prism 8 software (San Diego, CA, USA) was used to derive potencies (half maximal inhibitory concentrations or IC_50_s) and efficacies (maximal % growth inhibition) for each cell line.

### 2.9. ATP Assay

Total cellular ATP levels were measured using the CellTiter-Glo 2.0 Luminescent Cell Viability Assay (#G9241, Promega, Madison, WI, USA) according to manufacturer’s instructions. In brief, 1 × 10^4^ cells were seeded into 96-well black plates, incubated for 24 h and then treated with SHetA2 (5 or 10 µM) for 24 h. Then, the cells were incubated with CellTiter-Glo reagent, lysed and the luminescence signal was measured using a SYNERGY H1 microplate reader (BioTek, Winooski, VT, USA).

### 2.10. Mitochondrial-Stress and Glycolytic Rate Analysis

The metabolic profiles of endometrial cancer cells were assessed using XF Cell MitoStress Test and XF Glycolytic Rate Assay Kits (Seahorse Bioscience Inc., MA, USA) and the XFe96 Analyzer (Agilent, Santa Clara, CA, USA) per manufacturer’s instructions. In brief, XF cartridges were hydrated with sterile water overnight at 37 °C without CO_2_ (pH to 7.4 at 37 °C). The next day, cells grown on XFe96 tissue culture plates were treated with SHetA2 (10 µM) for 4 h and then the plates were washed with the DMEM medium and incubated at 37 °C in a non-CO_2_ incubator for 45 min. The sensor cartridge was hydrated with calibrant supplied with the kit at 37 °C without CO_2_ for 45 min before the start of the assay. Oligomycin (1 µM), FCCP (2 µM) and a mix of antimycin A and rotenone (0.5 µM each) were added into the appropriate ports of the cartridge and calibrated in the instrument. Then the cell culture plate was loaded in the instrument, and the assay was run per the standard template, and the oxygen consumption rate (OCR) was measured. For the Seahorse Glycolytic Rate Assay, the cartridge was loaded with mixture of antimycin A and rotenone (0.5 µM each) and 2-deoxy-D-glucose (2-DG, 50 mM) into the appropriate ports, and the experiment was performed as described above, except that the culture plate was washed again with assay medium before loading into the instrument. The OCR and proton efflux rates were calculated using the Seahorse XF Cell Mito Stress Report Generator and Seahorse XF Glycolytic Rate Assay Report Generator, respectively. Data were normalized to total protein concentration before plotting.

### 2.11. Annexin-V/PI Apoptosis Flow Cytometry Assay

Percentages of cells undergoing apoptosis were quantified by Annexin V and PI staining using the FITC Annexin V/Dead Cell Apoptosis Kit (# V13242, ThermoFisher, Waltham, MA, USA). In brief, endometrial cells treated with 10 µM SHetA2 for 24 h were collected, washed with cold PBS and then incubated with Annexin V-PI stain in the dark for 15 min at room temperature. Then the samples were subjected to flow cytometric analysis in accordance with the manufacturer’s guidelines.

### 2.12. Caspase-3 Activity Assay

Cells were treated with SHetA2 for 24 h and protein lysates were collected using cell lysis buffer mPER (#78501, ThermoFisher, Waltham, MA, USA). Using the BCA assay, cell lysates were diluted to 1 mg/mL concentration and caspase-3 activity was measured with the Caspase-3 Activity Assay Kit (#5723, Cell Signaling Technology, Danvers, MA, USA) according to manufacturer’s instructions. The relative fluorescence intensity was measured using a SYNERGY H1 microplate reader (BioTek, Winooski, VT, USA) at excitation and emission wavelength of 380 nm and 460 nm, respectively.

### 2.13. Mass Spectrometry Sample Preparation and LC-MS/MS Measurement

Ishikawa cells were treated with 10 µM of SHetA2 or solvent for 24 h in triplicate and protein isolates were collected in mPER. Samples were digested according to the filter aided sample preparation (FASP) protocol as described previously [47] and sent to the core lab facility at OUHSC for mass spectrometry analysis.

### 2.14. Co-Immunoprecipitation Assay

The interactions between mortalin and its client proteins in cultures treated with SHetA2 or vehicle for 4 h were studied by co-immunoprecipitation using the Pierce™ Co-Immunoprecipitation Kit followed by immunoblotting. For this, Ishikawa cells were treated with SHetA2 or vehicle for 4 h and protein isolates were collected. Approximately 500 μg of protein lysate was incubated with agarose beads (already coupled with 10 µg of mortalin antibodies per manufacturer’s protocol) overnight at 4 °C. Following washing with buffer provided in the kit, immuno-precipitated complexes were collected, re-suspended in sample buffer, and heated for 5 min at 95 °C. The co-immunoprecipitation of client proteins was detected by Western blot analysis using equal volumes of immuno-precipitated proteins.

### 2.15. Drug Interaction Analysis

Endometrial cancer cells were seeded in 96-well plates and treated with a twofold dilution series of SHetA2, Taxol or 1:1 combination of SHetA2 and paclitaxel in four replicates for 72 h followed by cell viability measurement using a MTT assay. The fold affect (Fa) was calculated by dividing the average ODs of the treated cultures by the average ODs of the controls and then subtracting this quotient from the number 1. Fold effects of the drugs were entered into CompuSyn Software (CompuSyn, New York, NY, USA) for production of isobolograms and derivation of combination indices (CIs) and Dose Reduction Indices (DRIs).

### 2.16. Tumor Xenograft Model

The animal study was approved by Institutional Animal Care and Use Committee (Protocol #18-118-CHI), University of Oklahoma Health Sciences Center and conducted in accordance with a standard animal protocol. For this study, female five-week-old SCID mice were purchased from Envigo Sales and allowed an acclimation period of 72 h. Then the animals were subcutaneously injection with 1 million Ishikawa cells, suspended in 100 µL sterile normal saline and the tumor size was monitored using calipers and formula ((width^2^ × length)/2), three times per week. The mice were randomized into four separate groups of 10 mice each so that there were no significant differences in tumor size between the groups as the start of treatment (ANOVA, *p* > 0.05). Treatment was initiated after the tumors reached an average volume of 50 mm^3^. Treatment groups consisted of controls placebo (30% Kolliphor HS 15 in water, daily), 60 mg/kg SHetA2 daily given by oral gavage, 10 mg/kg paclitaxel weekly administered by intraperitoneal (i.p.) injection, and the combination of 60 mg/kg SHetA2 daily with 10 mg/kg paclitaxel weekly. Mice were weighed once a week, and animal health was monitored daily. After 20 days of treatment, all mice were euthanized by controlled CO_2_ inhalation followed by cardiac puncture to assure death. Tumors were collected at necropsy and a portion of each tumor from each animal was fixed in paraformaldehyde and embedded in paraffin, while another portion was snap-frozen in liquid nitrogen. 

### 2.17. Immunofluorescence/Immunocytochemistry

Ishikawa cells (1 × 10^4^)-seeded chamber slides (Lab-Tek II) were treated with SHetA2 or DMSO for 24 h and stained with MitoTracker™ Red (CMXRos-M7512) for 30 min followed by PBS washing and fixation in 4% paraformaldehyde (pH 7.4) for 30 min at room temperature. Again, cells were washed with PBS (3×), permeabilized with 0.5% Triton X-100 in PBS for 5 min and incubated in 0.1% BSA in PBS for 30 min at room temperature. Then the cells were incubated with primary antibody for AIF or γH2AX (1:100, overnight at 4 °C), washed with PBS and incubated with the fluorochrome (Alexa Fluor 488)-tagged secondary antibody (1:500 dilution) for 1 h at room temperature. After washing with PBS, slides were mounted using ProLong Gold mounting medium with DAPI (#P36931, ThermoFisher, Waltham, MA, USA) and 1 mm thick cover glasses and left to dry overnight at room temperature and then imaged with a confocal microscope Leica SP2 (Leica, Wetzlar, Germany) at 63×.

### 2.18. Data and Statistical Analysis

All the experiments, unless otherwise stated, were performed in triplicate and repeated at three independent times. Data were expressed as means ± SD, unless otherwise noted. Student *t*-test and ANOVA were used to compare the means between two groups and among three or more groups, respectively. The Statistical significance was set at *p* < 0.05, using GraphPad Prism 8 software (GraphPad Software Inc., La Jolla, CA, USA). For mass spectrometry analysis, proteins with high confidence and <5% false discovery rate (FDR) were identified by using the two-stage step-up method of Benjamini, Krieger, and Yekutieli in GraphPad Prism Software. For the xenograft study, tumor volume at each time point was compared between the two treatment groups using a linear mixed-effects model for repeated measures.

## 3. Results

### 3.1. Cancerous Features

Our initial assessment of SHetA2 activity against endometrial cancers was to evaluate a range of physiologically-achievable uterine concentrations [36] on three human endometrial cancer cell lines (AN3CA, Hec1B, and Ishikawa) [48]. AN3CA and Hec1B do not harbor mutations in Grp78, hsc70, or mortalin, while Ishikawa has a F335L mutation in the Grp78 ATPase domain, a E600G in the Grp78 lid region, and a R99Q mutation in the mortalin ATPase domain, and no mutations in hsc70 [49]. In each of these cell lines, significant, dose-dependent reduction of anchorage-independent growth measured by colony forming ability in agarose, which is a surrogate for tumorigenic capacity in vivo (Figure 1A). SHetA2 also inhibited migration (Figure 1B) and invasion (Figure 1C) of all cell lines tested. Thus, the presence of Grp78 and mortalin mutations do not appear to interfere with SHetA2 activity.

### 3.2. Cell Cycle

Based on the support of the cyclin D1/cyclin dependent kinase (CDK)4/6 complex function by the SHetA2 targets, hsc70, and mortalin [48,49], we predicted that SHetA2 would cause G1 cell cycle arrest by disrupting this protection. To determine if SHetA2 inhibition of growth occurred in a specific portion of the cell cycle, cultures treated with 10 µM SHetA2 for 24 h were evaluated for PI staining using flow cytometry. As predicted, SHetA2 caused significant G1 cell cycle arrest in the Hec1B and Ishikawa cell lines, however it had no significant effect on the cell cycle profile of AN3CA (Figure 2A). Western blot analysis demonstrated that SHetA2 reduced levels of cyclin D1 only in the two cell lines that experienced G1 arrest (Figure 2B). Furthermore, SHetA2 elevated p21 and p27 in AN3CA and Ishikawa cell lines, while decreasing p21 and having no effect on p27 in Hec1B. These results demonstrate that SHetA2 induces G1 cell cycle arrest in association with reducing levels of the cyclin D1 G1-to-S promoter in two out of three cell lines, while altering levels of the p21 and p27 G1-to-S progression inhibitors to varying extents in different endometrial cancer cell lines.

### 3.3. Mitochondria and Metabolism

Based on SHetA2 disruption of mortalin complexes and the essential roles of mortalin in import of nuclear encoded proteins into, and maintenance of, mitochondria [50], we predicted that SHetA2 would cause mitochondrial damage and altered metabolism in endometrial cancer cells. Consistent with this prediction, SHetA2 decreased mitochondrial membrane potential as indicated by significant reduction of the ratio of JC-1 dye aggregates to monomers in all three cell lines treated with SHetA2 (Figure 3A). Moreover, SHetA2 significantly decreased levels of mitochondrial proteins involved in fusion, namely long form of outcome predictor in acute leukemia 1 (OPA1L), mitofusion 1 (MFN1), mitofusion2 (MFN2), and fission, namely, phosphorylated dynamin-related protein 1 (pDrp1) in the endometrial cancer cell lines (Figure 3B). The OPA1S isoform, which is dispensable for fission under stressed conditions, was not affected by SHetA2 [51]. These results demonstrate that SHetA2 damages mitochondria, which could interfere with oxidative phosphorylation (OxPhos) known to occur in the mitochondria.

Next, we evaluated SHetA2 effects on metabolism in the mitochondria (OxPhos) and cytoplasm (glycolysis). SHetA2 dose-dependently reduced mitochondrial metabolic viability measured with the MTT mitochondrial-metabolized dye in all three cell lines (Figure 3C). Consistent with this decrease, SHetA2 also dose-dependently reduced total cellular levels of ATP in each of the three cell lines (Figure 3D). Results from a Seahorse Mito Stress Test indicated that SHetA2 blocked both basal and induced levels of OxPhos in the Ishikawa cell line (Figure 3E) and put cells into a quiescent metabolic state (Figure 3F). Data from a Seahorse Glycolysis Rate Assay indicated that SHetA2 also inhibited both basal and induced glycolysis (Figure 3G) and put cells into a quiescent state (Figure 3H). These results demonstrate that SHetA2 inhibits both OxPhos and glycolysis in endometrial cancer cell lines.

### 3.4. Mechanisms of SHetA2

To evaluate the mechanism of SHetA2 action against endometrial cancer cell lines, we used mass spec analysis to identify proteins differentially expressed in Ishikawa cells treated with 10 µM SHetA2 for 24 h, compared to control cultures treated with solvent only. Forty-eight proteins were found to be significantly altered in the treated compared to the untreated cultures (<5% FDR, Appendix A). Ingenuity analysis identified four networks of interacting proteins among the significantly regulated proteins (Appendix A). The top network consists of 18 proteins that include two of the SHetA2 targets, Grp78 (HSPA5) and hsc70 (HSPA8) (Appendix A). Ingenuity Analysis identified the top 4 canonical pathways, in which the SHetA2-regulated proteins are involved, to be Eukaryotic Initiation Factor 2 (EIF2) signaling (8 proteins, *p* = 1.8 × 1^−8^), glycolysis (4 proteins, *p* = 2.34 × 10^−7^), ER stress (5 proteins, 1.08 × 10^−5^) and gluconeogenesis (3 proteins, 2.1 × 10^−5^). The associations of the individual proteins in these pathways are listed in Appendix A. In summary, the major effect of SHetA2 on Ishikawa cells involves regulation of Grp78 and hsc70 and their down-stream targets involved in regulation of ER stress, metabolism and protein synthesis.

In addition to altering protein levels, SHetA2 can affect the functions of specific client proteins by disrupting their binding to the mortalin chaperone [17,25]. Therefore, we predicted that some of the client proteins released from mortalin by SHetA2 would be metabolic enzymes. A screen of five metabolic enzymes found to be affected by SHetA2 in other cancers, demonstrated that SHetA2 reduced mortalin co-immunoprecipitation of aldehyde dehydrogenase 18 family member A1 (ALDH18A1), cytidine triphosphate synthetase (CTPS), malate dehydrogenase (MDH1), and enoyl Coenzyme A hydratase, short chain, 1, mitochondrial, (ECHS1) in Ishikawa cells (Figure 4A). SHetA2 also disrupted binding of mortalin to the inositol trisphosphate receptor (IP3R), which is an essential interaction for import of calcium into the mitochondria [52]. Total cellular levels of IP3R were also reduced in SHetA2-treated Ishikawa cells (Figure 4B). Taken together, these findings suggest that the mechanism of SHetA2 induction of mitochondrial damage and altered metabolism are associated with disruption of mortalin/client protein complexes.

We used Western blot analysis to monitor biomarkers of cell pathway responses to SHetA2 treatment (Figure 4B). The endometrial cancer cell lines appear to respond to SHetA2 by elevating autophagy, a cell recycling pathway, as indicated by increased microtubule-associated proteins Light Chain 3 (LC3) II isoform [53] (Figure 4B). This autophagy may not be blocked as indicated by elevated p65 protein [53] (Figure 4B). The ultimate effect of SHetA2 treatment, however, appears to culminate in apoptotic cell death as documented by elevation of the numbers of Annexin V- and PI-stained cells in treated cultures (Figure 4C). Western blot analysis confirmed that SHetA2-induced cell death is associated with cleavage of poly (ADP-ribose) polymerase (PARP) and AIF, and reduced levels of total caspase 3 (Figure 4D). We used an ELISA assay to confirm dose-responsive activation of caspase 3 in all three cell lines. Cleavage of caspase 3 is known to activate its proteolytic activity, which cleaves and thereby inactivates PARP [54]. Cleavage of AIF from its membrane-bound form in mitochondria causes its relocation to the nucleus where it promotes DNA damage [55]. DNA damage in SHetA2-treated cells was confirmed by elevation of histone variant γH2AX, which is a biomarker of DNA double-stranded breaks [56] (Figure 4D). Thus, SHetA2 causes death of endometrial cancer cells through multiple mechanisms including caspase- and AIF-mediated apoptosis.

### 3.5. Synergy with Paclitaxel in Cell Culture and In Vivo

To test the feasibility of combining SHetA2 with paclitaxel in treating endometrial cancers, we derived the CI’s of this drug combination using isobologram analysis according to the Chou–Talalay method [57]. SHetA2 and paclitaxel acted synergistically across all fold affect (Fa) combination doses above 0.1 in all three cell lines (Figure 5A). The DRIs derived from these analyses predict that when SHetA2 is combined paclitaxel, the dose of paclitaxel can be reduced by 6.4-fold for AN3CA, 2.4-fold for AN3CA, and 3.5-fold for Ishikawa to achieve the same 0.90 Fa. Given these promising results, we then evaluated the drug combination in mice harboring Ishikawa xenografts and found that each drug as a single agent significantly reduced tumor growth, while the combination was significantly more effective than either drug alone (Figure 5B). Furthermore, none of the drug treatments altered the body weights of the animals in comparison to vehicle treated controls (Figure 5C). These results indicate that SHetA2 and paclitaxel can be combined to increase efficacy without increasing toxicity in an endometrial cancer xenograft model.

To evaluate the mechanism of SHetA2 and paclitaxel interaction, we evaluated AIF truncation by Western blot (Figure 6A). Paclitaxel did not cause AIF truncation as a single agent, nor did it prevent SHetA2-induced AIF truncation. To further verify the role of AIF in SHetA2-induced cell death, we used fluorescence microscopy to image AIF cellular localization (Figure 6B) and γH2AX nuclear staining (Figure 6C), which confirmed that SHetA2 induced AIF translocation to the nucleus and DNA damage, respectively. Although paclitaxel did not induce AIF translocation, it did cause DNA damage as a single agent. In summary, these results indicate that both drugs cause DNA damage, and that the mechanism of SHetA2 and paclitaxel synergy does not include enhancement of AIF translocation.

## 4. Discussion

The results of this study demonstrate the feasibility of developing the novel SHetA2 drug for treatment of endometrial cancers. The anti-cancer activities of SHetA2 include inhibition of anchorage-independent growth, migration, invasion, and G1-S cell cycle progression. The original rationale for using this drug to target Grp78, hsc70, and mortalin is supported by proteomic analysis demonstrating that SHetA2 alters Grp78, hsc70, and their client proteins, and also disrupts mortalin/client protein complexes in endometrial cancer cells. Inherent mutations of these proteins in Ishikawa cells did not prevent SHetA2 activity. Down-stream consequences of disrupting mortalin support of mitochondria in endometrial cancer cells was observed as reduction of mitochondrial membrane potential and mitochondrial proteins involved in fission and fusion. Furthermore, both mitochondrial OxPhos and cytoplasmic glycolysis were reduced by SHetA2 putting endometrial cancer cells into quiescent metabolic states. Although the cells appeared to respond by upregulation of autophagy, this was not sufficient to prevent the ultimate cell death through a caspase- and AIF-dependent mechanism.

Combination treatment of SHetA2 with paclitaxel was found to be synergistic in each of the three cell lines tested. Evaluation of this combination in an in vivo Ishikawa xenograft model confirmed that the combination was more effective than either drug alone, and acted without added observable toxicity. While both drugs induced nuclear DNA double-stranded breaks as indicated by γH2AX staining, their mechanisms leading to this damage were different. The damage caused by SHetA2 to mitochondria was associated with cleavage and release of AIF to translocate to the nucleus, where it is known to promote DNA damage [55]. On the other hand, elevated nuclear γH2AX staining was not associated with AIF cleavage or nuclear translocation in paclitaxel-treated cells.

The lack of significant toxicity observed for SHetA2 in extensive preclinical studies supports the potential for this drug to not cause added toxicity when combined with paclitaxel. This is supported by lack of body weight changes in our drug combination endometrial cancer xenograft model. Paclitaxel causes significant side effects in endometrial cancer patients, the worst of which is neuropathy [58]. The DRIs derived in our Cho–Talalay analysis predict that paclitaxel could be reduced 2- to 6-fold in patients also receiving SHetA2 to achieve the same cancer control. Thus, the potential of SHetA2 to allow lower doses of paclitaxel treatment while not inducing toxicity on its own, indicate high potential of this drug for improving the therapeutic ratio of paclitaxel-based therapy.

## 5. Conclusions

In summary, the results of this preclinical study support the further development of SHetA2 as a single agent, and in combination with paclitaxel, for treatment of endometrial cancer. This treatment has the potential to reduce treatment-related side effects. Further study has the potential to identify molecular factors, such as SHetA2-target protein levels and mutations in their client proteins, which could be used to predict which patients would most likely benefit from SHetA2-based therapy.

## Figures and Tables

**Figure 1 cancers-13-02322-f001:**
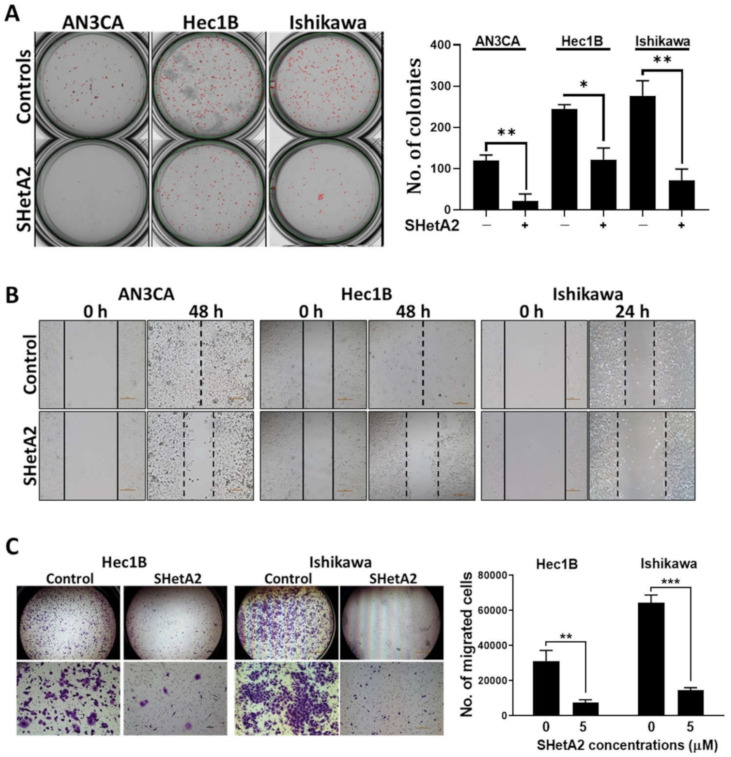
SHetA2 attenuates cancerous phenotypes of endometrial cancer cell lines: (**A**) Soft agar colony formation assay of endometrial cancer cells treated with 5 µM SHetA2 or vehicle. After 3 weeks of culture, colonies were imaged (left panel) and counted (right panel) using q GelCount colony counter. Representative images scanned at 4× magnification are shown. The controls were set as 100% and data are shown as mean ± SD. (**B**) Wound healing scratch assay of endometrial cancer cells treated with 5 µM SHetA2 or vehicle. Images were captured at the indicated time points. (**C**) Invasion assays were conducted using the Matrigel-coated transwell chamber. Representative images of transwells showing invasion capacity of Hec1B and Ishikawa endometrial cancer cell in the presence of SHetA2 or vehicle are shown (left panel). The number of invaded cells were counted with a cell counter and shown (right panel). * *p* ≤ 0.05, ** *p* ≤ 0.01, *** *p* ≤ 0.001 when compared with respective controls.

**Figure 2 cancers-13-02322-f002:**
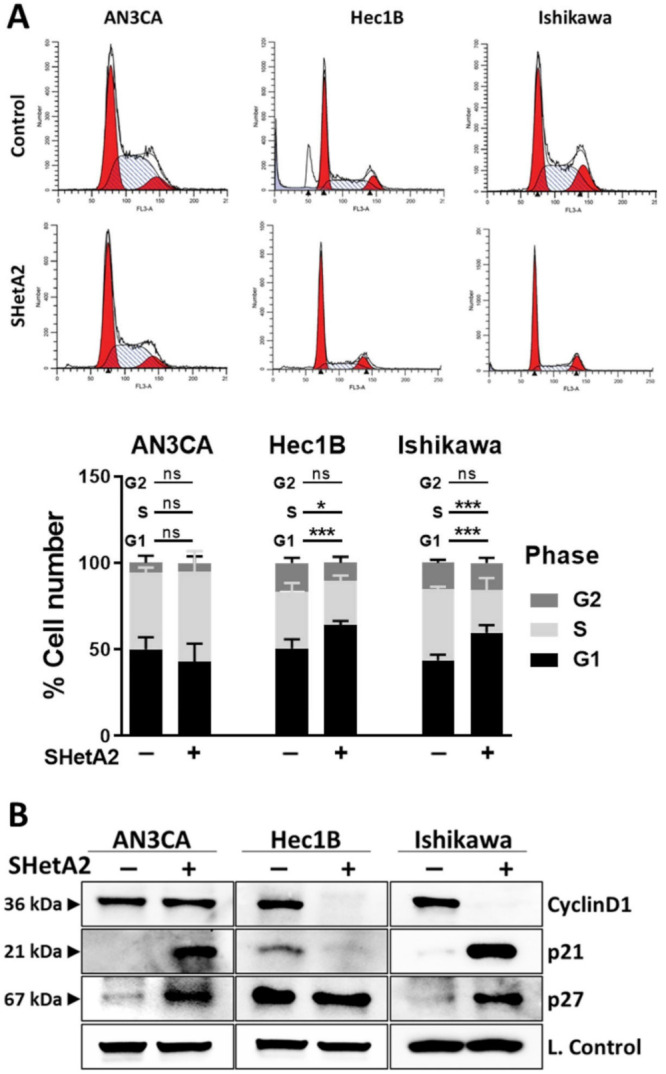
SHetA2 induces cell cycle arrest and alters expression of cell cycle regulatory proteins: (**A**) Flow cytometry analysis of endometrial cancer cells treated with SHetA2 (10 µM) or vehicle for 24 h followed by PI staining. Representative images of cell cycle distributions are shown (upper panel). Percentages of the cell population at different cell cycle phases were calculated from three independent experiments and shown in the bar graph (Lower panel). (**B**) Western blot analysis of cell cycle regulatory proteins isolated from endometrial cancer cells treated with SHetA2 (10 µM) or vehicle for 24 h. * *p* ≤ 0.05, *** *p* ≤ 0.001 when compared with respective controls.

**Figure 3 cancers-13-02322-f003:**
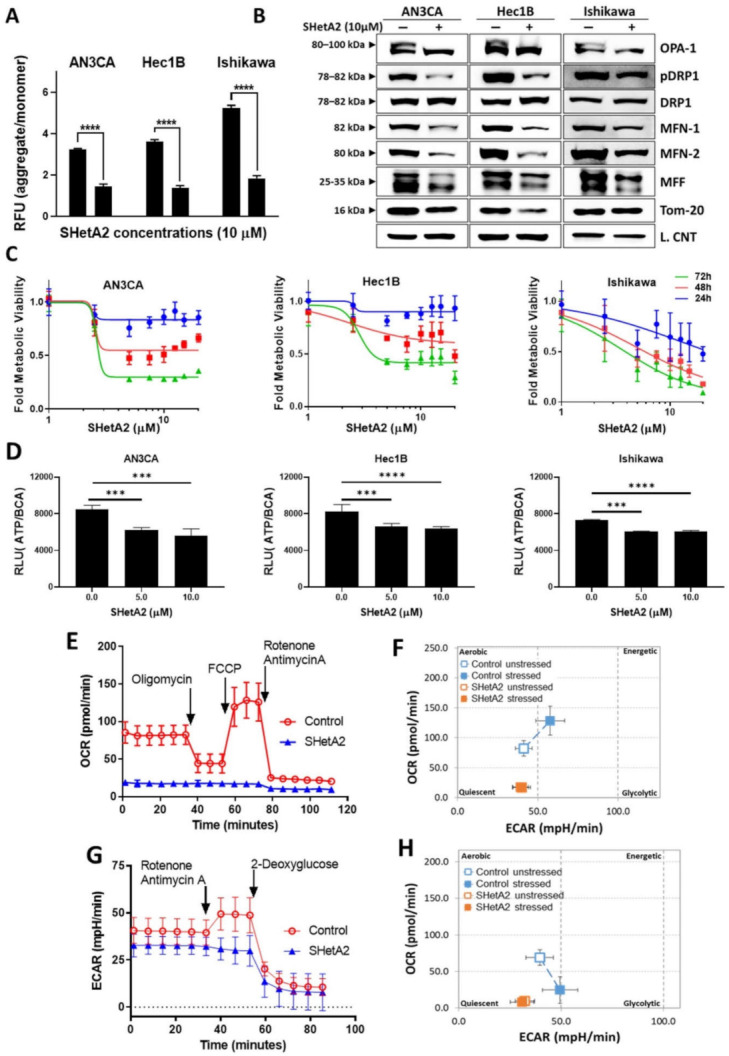
SHetA2 affects endometrial cancer cell mitochondria and metabolism: (**A**) JC-1 assay (ratio of aggregated JC-1 to monomer JC-1) in endometrial cancer cells treated with SHetA2 (10 µM) or vehicle. (**B**) Western blot analysis of mitochondrial dynamics-related proteins in endometrial cancer cells treated with or without SHetA2 (10 µM) for 24 h. (**C**) Metabolic viability of endometrial cancer cells treated with various doses of SHetA2 for 24, 48, and 72 h. The results are shown as % viability as compared to control. (**D**) ATP assay of endometrial cancer cells treated with SHetA2 (5 and 10 µM) for 24 h. (**E**) OCR and (**F**) Energy phenotype of Ishikawa cells treated with SHetA2 (10 µM) or vehicle for 4 h and measured by the Seahorse Cell Mito-Stress Test and the XFe96 Analyzer. (**G**) Extracellular acidification rate (ECAR) and (**H**) energy phenotype of Ishikawa cells treated with SHetA2 (10 µM) or vehicle for 4 h and measured with the Seahorse glycolytic rate assay and the XFe96 Analyzer. *** *p* ≤ 0.001, **** *p* ≤ 0.0001 when compared with respective control.

**Figure 4 cancers-13-02322-f004:**
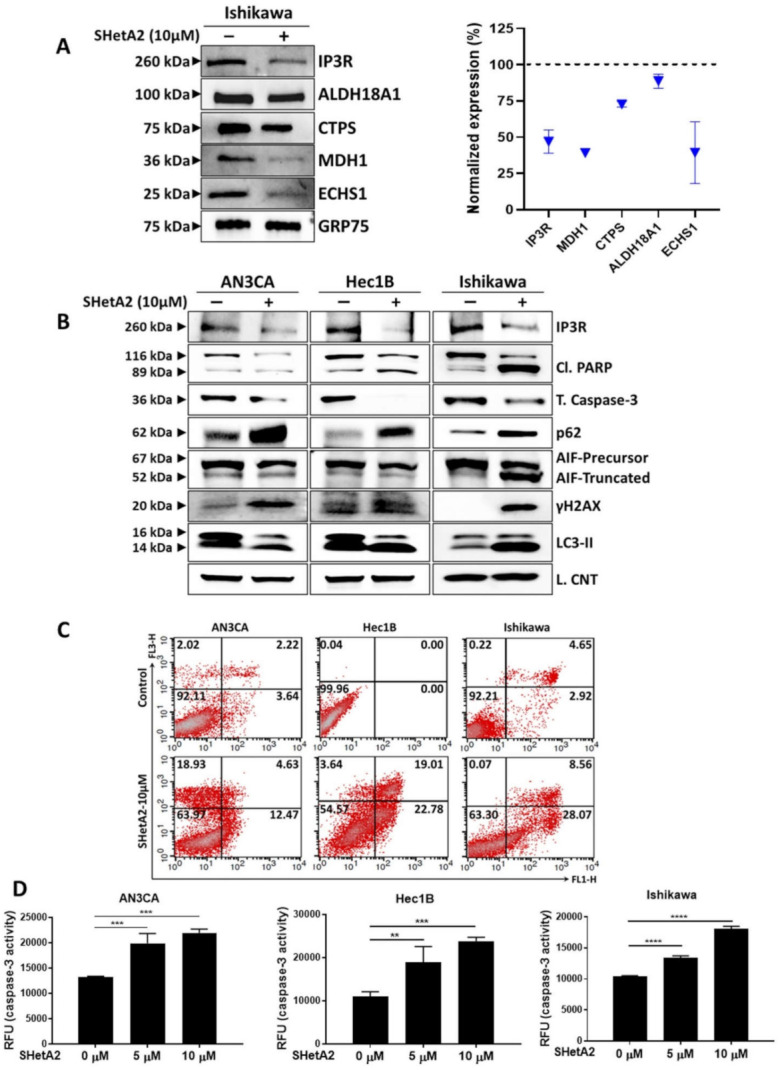
Molecular mechanism of SHetA2: (**A**) Ishikawa cells were treated with SHetA2 (10 µM) for 4 h and co-immunoprecipitation was performed with mortalin antibody (Grp75)-tagged beads. Western blots (left panel) and densitometry analyses (right panel) was performed for IP3R, ALDH18A1, CTPS, MDH1, and ECHS1 to confirm the reduction of client protein co-immunoprecipitation. (**B**) Western blot analysis for the protein expression of IP3R, Cl. PARP, total caspase-3, P62, AIF, DNA damage marker (γH2AX) and LC3-II proteins in endometrial cancer cells treated with, or without, SHetA2 (10 µM) for 24 h. (**C**) Cell apoptosis was detected by Annexin-V/PI combined labeling with flow cytometry in endometrial cancer cells 24 h after treatment with SHetA2 (10 µM) or vehicle. (**D**) Caspase-3 activity assay for endometrial cancer cells treated with indicated dose of SHetA2 for 24 h. ** *p* ≤ 0.01, *** *p* ≤ 0.001, **** *p* ≤ 0.0001 when compared with respective control.

**Figure 5 cancers-13-02322-f005:**
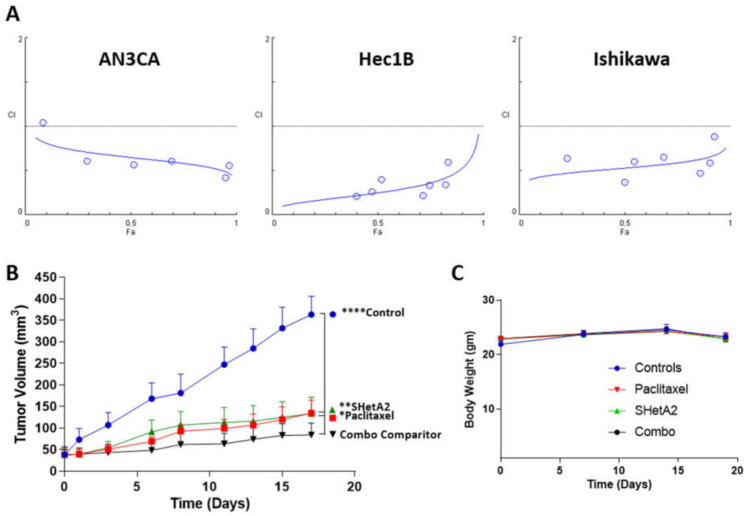
SHetA2 enhances paclitaxel activity without toxicity: (**A**) CIs for endometrial cancer cell lines treated with a series of 2-fold dilutions of SHetA2 and paclitaxel in their 1:1 ratio of their IC_50_ concentrations in parallel with single drug treatments. After 72 h, the MTT assay was used to determine theFa’s, and CompuSyn was used to determine the CI’s. CI points below the dotted line indicate synergy. (**B**,**C**) Mice bearing Ishikawa-endometrial cancer xenografts were treated with oral 60 mg/kg/day SHetA2, 10 mg/kg/week i.p. paclitaxel, combination, or vehicle control. Average tumor volumes (**B**) and body weights (**C**) throughout the study are shown. ANOVA comparison of all treatment groups to the SHetA2 plus paclitaxel treated group (Combo comparator) * *p* < 0.05, ** *p* < 0.01 and **** *p* < 0.0001.

**Figure 6 cancers-13-02322-f006:**
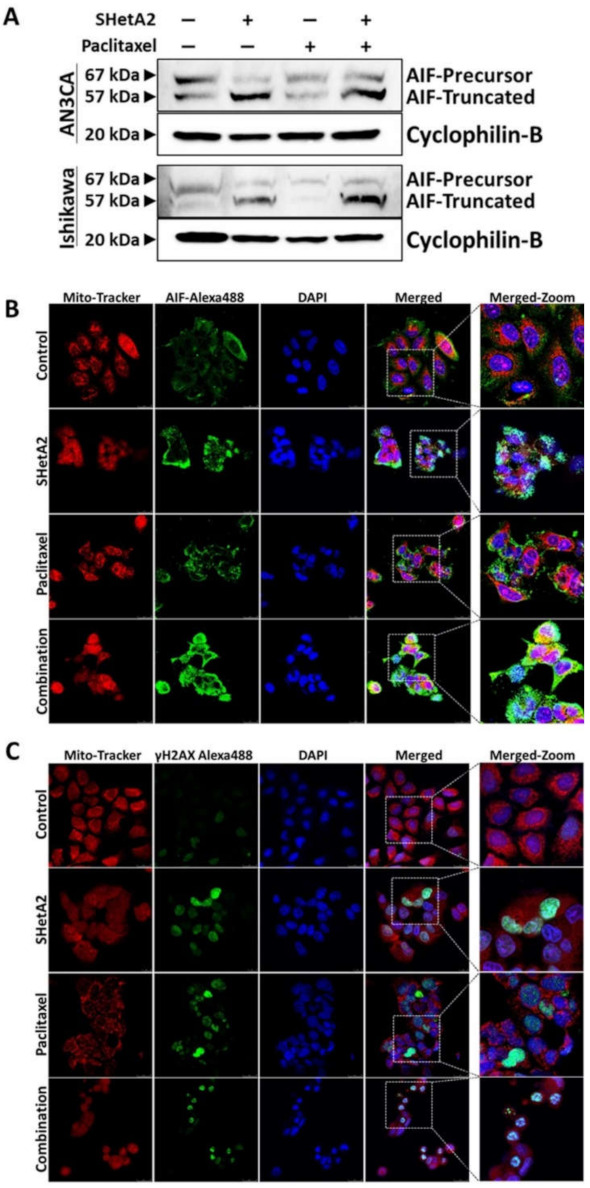
SHetA2 induces nuclear translocation of AIF, while both SHetA2 and paclitaxel induce DNA damage: (**A**) Western blot analysis for AIF expression in AN3CA and Ishikawa cells treated with SHetA2 or paclitaxel or their combination. (**B**,**C**) Representative immunofluorescence images for Mitotracker (red) and AIF (Green, **B**), or γH2AX (**C**) show their localization in AN3CA and Ishikawa cells treated with SHetA2 or paclitaxel or their combination. DAPI (Blue) staining was used as nuclear stain. Representative images taken at 63× magnification are shown. The left-most panels represent enlarged pictures of the square boxes shown in merged image.

## Data Availability

Proteomic data generated in this study is available in the Appendix A.

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
