# Peer review of "Utility and Mechanism of SHetA2 and Paclitaxel for Treatment of Endometrial Cancer"

_cancers, 2021, doi:10.3390/cancers13102322_

Round 1
Reviewer 1 Report
The authors should be commended on a sound preclinical study of the novel compound SHetA2 in endometrial cancers. The methodology and results are nicely presented, and the findings reported support further development of Phase 1 clinical trials.
Author Response
We thank Reviewer 1 for the praise of our work and recognizing the support it provides for translation of SHetA2 into clinical trial.
Reviewer 2 Report
Utility and mechanisms of SHetA2 and paclitaxel for treatment of Type I endometrial cancers
Reviewer report
Thank you for the opportunity to review this paper. This paper investigated a novel compound, SHetA2, in the treatment of endometrial cancer. SHetA2 is a Sulfur Heteroarotinoid that binds to Grp78, Hsc70 and mortalin, disrupting cell survival.
The authors have completed a significant number of mechanistic experiments in 3 cell line models AN3CA, Ishikawa and HEC1B. They then looked at the drug in combination with paclitaxel in a mouse model. The results are promising in that combination SHetA2/Pac significantly decreased tumour size without additional toxicity.
However, there are a few major issues I have with the study:
- Personally, I find the use of ‘type I/type II’ endometrial cancer outdated and irrelevant. Whilst used traditionally (introduced in 1983!), molecular classification has shown that actually there are more than 2 types and describing tumours based on histology/grade/stage is more clinically appropriate (maybe we will move to molecular classification sometime in the future…).
So here you are describing type I EC at high risk of recurrence – I think you mean the population of women with endometrioid endometrial cancer with stage 1 disease (grade 3, >50% myometrial invasion, LVSI) or stage II-III. Furthermore you state that paclitaxel is the standard of care for type I stage I endometrial cancer at high risk of recurrence – in fact, most stage I recurrences are loco-regional in the vaginal vault and therefore high risk stage I cases would receive brachytherapy.
I think just framing your introduction would help; also, why can’t SHetA2 also be used for serous endometrial cancer (type II?) as these are high risk and more often treated with chemotherapy? They would be a group that would benefit from this treatment.
- Does SHetA2 activity rely on the tumour to have Grp78, Hsc70 or mortalin mutations? How does this relate to your cell lines – do they have mutations in these genes? Will this affect your results? Whilst endometrial cancer does have one of the highest mutation frequencies of these genes, they do only occur in about 10% of all cases.
- The objective of the study was to evaluate the utility and mechanisms of SHetA2 as a single agent and in combination with paclitaxel for type I endometrial cancer. I think you answered the single agent question well, however I felt there was information lacking around combination therapy – to me this is the most interesting/exciting part (and highlighted in your title), however you only showed one mouse model/cell line (figure 5b). The paper would benefit from additional models, and/or investigating the mechanisms of action via fluorescent microscopy in the mouse tissue samples, rather then going back to cells (fig 6).
Minor comments:
- Please list all SHetA2 concentrations used in the methods (for example it is omitted in colony formation assay, invasion assay, MMP assay). In some methods they are treated with 5uM, and some with 10uM. Why is this not consistent?
- Figure 1: If SHetA2 has a significant effect on cell viability, this will effect migration/invasion/colony formation. It seems like the results are such because SHetA2 kills off most of the cells. Therefore, rather then measuring migration, you are just measuring lack of cells to migrate.
It would be important to include a simple cell viability/proliferation assay to show this, or use proliferation inhibitors for the migration/invasion assay?
- I was confused by the western blots in figure 2, especially for Hec1B. I thought the blots did not reflect the cell cycle distributions, and SHetA2 had different effects on the 3 cell types.
- For experiments where only data for ishikawa cells is presented, is there data for the 2 other cell lines?
- Mouse model: the use of paclitaxel at the same concentration as it is used in combination with SHetA2 may not be relevant – here I think the clinically relevant amount should be used, as you want to show that SHetA2+Pac has the same/better anti-tumour effect than the current treatment, but without the toxicity.
Addition of the mouse model with other cell lines would benefit the paper.
Investigation into combination effects in the tumour tissue would benefit the paper.
- Over all there are grammatical and spelling errors – for example results 3.6: fold reduction of paclitaxel “AN3CA, HN3CA and Ishikawa’ should be AN3CA, HEC1B and Ishikawa’. Please check manuscript thoroughly.
Author Response
Response to Reviewer 2: We thank this reviewer for the thoughtful critique.
However, there are a few major issues I have with the study:
- Concerns regarding focus of introduction on type1/type2 endometrial cancer, because it is an outdated term and “…why can’t SHetA2 also be used for serous endometrial cancer..?”.
We agree that type1/type2 classification is outdated and have removed these details from the manuscript. We presented the type1/type2 classification in the introduction because the three cell types that we used in the study presented are classified as Type 1. Furthermore, we did not have access to serous endometrial cancer cell lines, since serous endometrial cancer cell lines are not commercially available. Recently, our university signed a material transfer agreement that allowed us to purchase serous endometrial cancer cell lines from Yale University. We are now generating data with those cell lines and intend to submit a distinct manuscript focused on serous endometrial cancer once we have sufficient data and new findings.
- Does SHetA2 activity rely on the tumour to have Grp78, Hsc70 or mortalin mutations? How does this relate to your cell lines – do they have mutations in these genes? Will this affect your results? Whilst endometrial cancer does have one of the highest mutation frequencies of these genes, they do only occur in about 10% of all cases.
We used the Broad Institute Using the Broad Institute Cancer Cell Line Encyclopedia (https://portals.broadinstitute.org/ccle) to search for Grp78, Hsc70 or mortalin mutations in the cell lines used. We presented this data in the results section on lines 324 to 327:
AN3CA and Hec1B do not harbor mutations in Grp78, hsc70 or mortalin, while Ishikawa has a a F335L mutation in the Grp78 ATPase domain, a E600G in the Grp78 lid region, and a R99Q mutation in the mortalin ATPase domain, and no mutations in hsc70 [47].
And on lines 330-331:
Thus, the presence of Grp78 and mortalin mutations do not appear to interfere with SHetA2 activity.
And in the discussion on lines 506-507: Inherent mutations of these proteins in Ishikawa cells did not prevent SHetA2 activity.
- The objective of the study was to evaluate the utility and mechanisms of SHetA2 as a single agent and in combination with paclitaxel for type I endometrial cancer. I think you answered the single agent question well, however I felt there was information lacking around combination therapy – to me this is the most interesting/exciting part (and highlighted in your title), however you only showed one mouse model/cell line (figure 5b). The paper would benefit from additional models, and/or investigating the mechanisms of action via fluorescent microscopy in the mouse tissue samples, rather then going back to cells (fig 6).
We agree that the additional studies suggested would strengthen the manuscript, however due to our limited resources for conducting additional studies, we hope to be able to proceed to publication with the demonstration of synergy in three cell lines in culture and in one cell line xenograft model.
Minor comments:
- Please list all SHetA2 concentrations used in the methods (for example it is omitted in colony formation assay, invasion assay, MMP assay). In some methods they are treated with 5uM, and some with 10uM. Why is this not consistent?
We have carefully gone through the manuscript and assured that all methods accurately describe the concentration of SHetA2 used. In some experiments, we reduced the concentration of SHetA2 from 10 to 5 µM, because the high lethality of the 10 µM dose precluded our ability to accurately measure endpoints not associated with cell death.
- Figure 1: If SHetA2 has a significant effect on cell viability, this will effect migration/invasion/colony formation. It seems like the results are such because SHetA2 kills off most of the cells. Therefore, rather then measuring migration, you are just measuring lack of cells to migrate.
It would be important to include a simple cell viability/proliferation assay to show this, or use proliferation inhibitors for the migration/invasion assay?
We include cell viability assays in Figure 3C. This data shows that the dose and time used (5 µM for 48 hours) reduces viable cells by only 50%, however most of the effect sizes in Figure 1 are greater than 50%.
- I was confused by the western blots in figure 2, especially for Hec1B. I thought the blots did not reflect the cell cycle distributions, and SHetA2 had different effects on the 3 cell types.
The description of results on lines 347 to 358 was corrected to more accurately reflect the data as pasted below:
As predicted, SHetA2 caused significant G1 cell cycle arrest in the Hec1B and Ishikawa cell lines, however it had no significant effect on the cell cycle profile of AN3CA (Figure 2A). Western blot analysis demonstrated that SHetA2 reduced levels of cyclin D1 only in the two cell lines that experienced G1 arrest (Figure 2 B). Furthermore, SHetA2 elevated p21 and p27 in AN3CA and Ishikawa cell lines, while decreasing p21 and having no effect on p27 in Hec1B. These results demonstrate that SHetA2 induces G1 cell cycle arrest in association with reducing levels of the cyclin D1 G1-to-S promoter in two out of three cell lines, while altering levels of the p21 and p27 G1-to-S progression inhibitors to varying extents in different endometrial cancer cell lines.
- For experiments where only data for ishikawa cells is presented, is there data for the 2 other cell lines?
Each of our studies evaluated all three cell lines, except for metabolism and xenograft studies, which were limited to the Ishikawa cell line, because we did not have the resources to include the other two cell lines in these more extensive and expensive experimental approaches. We do not have data on AN3CA and Hec1B for the metabolism and xenograft experiments.
- Mouse model: the use of paclitaxel at the same concentration as it is used in combination with SHetA2 may not be relevant – here I think the clinically relevant amount should be used, as you want to show that SHetA2+Pac has the same/better anti-tumour effect than the current treatment, but without the toxicity.
Addition of the mouse model with other cell lines would benefit the paper.
Investigation into combination effects in the tumour tissue would benefit the paper.
We agree with this comment and believe that further study of varying doses of paclitaxel with SHetA2 are warranted based on this preliminary data. We chose the dose of paclitaxel used in our first experiment based on extensive use of this dose in the published literature. Also, we agree that an additional mouse model and studies of the tumor tissue would benefit the paper, however we do not have this additional experimental data to include in this submission.
- Over all there are grammatical and spelling errors – for example results 3.6: fold reduction of paclitaxel “AN3CA, HN3CA and Ishikawa’ should be AN3CA, HEC1B and Ishikawa’. Please check manuscript thoroughly.
We have carefully reviewed the manuscript and corrected grammatical and spelling errors.
Reviewer 3 Report
Title: Utility and mechanism of SHetA2 and paclitaxel for treatment of Type I endometrial cancers
The authors demonstrated SHetA2 inhibited anchorage-independent growth, migration, invasion and ATP production in endometrial cancer cell lines. Also, SHetA2 and paclitaxel exhibited synergistic combination anti-cancer effect in endometrial cancer. This study is well organized and contains some novel findings.
Minor comment
If the authors could assess SHetA2 in the paclitaxel resistance cell line, it would be perfect.
Author Response
We thank Reviewer 3 for his praise of our work and agree with his minor comment that evaluation of SHetA2 in a paclitaxel resistant cell line would be perfect. We have not been able to find commercially available taxol-resistant endometrial cancer cell lines for this study. Therefore, we are in the process of generating our own taxol-resistant sublines for future studies.
Round 2
Reviewer 2 Report
I thank the authors for addressing my comments, and believe the manuscript is acceptable for publication.
I look forward to reading your research in serous cell lines and suggest using primary patient derived cell lines rather than commercial? As you mentioned, there are not many commercial available.